# Interaction of insecticidal proteins from *Pseudomonas* spp. and *Bacillus thuringiensis* for boll weevil management

**Jardel Diego Barbosa Rodrigues**[◉][*], **Raquel Oliveira Moreira**[‡], **Jackson Antônio Marcondes de Souza**[‡], **Janete Apparecida Desidério**[◉]

Biology Department, Faculty of Agrarian and Veterinary Sciences (Jaboticabal Campus), São Paulo State University (UNESP), São Paulo, Brazil

◉ These authors contributed equally to this work.
‡ ROM and JAMS also contributed equally to this work.
* jardel.diego@unesp.br

**Data Availability Statement:** All relevant data are within the paper and its Supporting information files.

## Abstract

Cotton crop yields are largely affected by infestations of *Anthonomus grandis*, which is its main pest. Although *Bacillus thuringiensis* (Bt) derived proteins can limit insect pest infestations, the diverse use of control methods becomes a viable alternative in order to prolong the use of technology in the field. One of the alternative methods to Bt technology has been the utilization of certain *Pseudomonas* species highly efficient in controlling coleopteran insects have been used to produce highly toxic insecticidal proteins. This study aimed to evaluate the toxicity of IPD072Aa and PIP-47Aa proteins, isolated from *Pseudomonas* spp., in interaction with Cry1Ia10, Cry3Aa, and Cry8B proteins isolated from *B. thuringiensis*, to control *A. grandis* in cotton crops. The genes IPD072Aa and PIP-47Aa were synthesized and cloned into a pET-SUMO expression vector. Moreover, Cry1Ia10, Cry3Aa, and Cry8B proteins were obtained by inducing recombinant *E. coli* clones, which were previously acquired by our research group from the Laboratory of Bacteria Genetics and Applied Biotechnology (LGBBA). These proteins were visualized in SDS-PAGE, quantified, and incorporated into an artificial diet to estimate their lethal concentrations (LC) through individual or combined bioassays. The results of individual toxicity revealed that IPD072Aa, PIP-47Aa, Cry1Ia10, Cry3Aa, and Cry8B were efficient in controlling *A. grandis*, with the latter being the most toxic. Regarding interaction assays, a high synergistic interaction was observed between Cry1Ia10 and Cry3Aa. All interactions involving Cry3Aa and PIP-47Aa, when combined with other proteins, showed a clear synergistic effect. Our findings highlighted that the tested proteins in combination, for the most part, increase toxicity against *A. grandis* neonate larvae, suggesting possible constructions for pyramiding cotton plants to the manage and the control boll weevils.

## Introduction

Cotton is among the world's most important fiber crops. The 100 cotton-producing countries collectively planted approximately 33 million hectares of cotton in the 2021/2022 crop season

**Funding:** The authors declare that there is no financial disclosure.

**Competing interests:** The authors have declared that no competing interests exist.

and produced around 27 million tons of lint, driving an extensive supply chain and commercial activities involving domestic supply, imports, and exports [1, 2]. The ranking of the top four world producers includes India, China, the USA and Brazil, accounting for about 75% of global production [2, 3]. Brazil ranks as the 4th largest global producer and exports approximately 71% of its production [1, 4] and according to the Brazilian Institute of Geography and Statistics [5], in the 2021/2022 crop year, the country reached about 1.53 million hectares of cotton cultivation, representing a growth of 10.9% compared to the 1.38 million hectares grown in the previous crop year.

Despite cotton being a crop with significant potential for expansion due to its adaptability to the soils and climates of the Americas, the crop's productivity is affected by the attack of *Anthonomus grandis*, which is currently the most important pest in cotton farming in the Western Hemisphere [6]. In conventional cotton cultivation, chemical control remains one of the most widely used methods. The excessive use of these compounds contributes to increased production costs, which can represent up to 25% [7], as well as environmental damage. For these reasons, new strategies and alternative methods have been increasingly developed for the biological control of agricultural pests, aiming to decrease the volume of pesticides released into the environment. Thus, the use of biological insecticides and genetically-modified (GM) plants is advancing in the global market and gaining increasing space in farming.

The most used bacterium in controlling numerous insect pests is the species *Bacillus thuringiensis* [8]. Its main characteristic is the production of crystalline inclusions of protein nature during the sporulation phase, which is called Cry proteins or δ-endotoxins [9]. These proteins are highly specific to target insects and form the basis of different products used as biological insecticides [10]. Upon ingestion of these proteins, the high intestinal pH of susceptible lepidopteran insect larvae activates the protoxin that binds to specific receptors, inducing lesions that destroy the intestinal cells and lead insects to death by starvation [9]. When it comes to insects of the coleopteran order, some of the Bt toxins with potential activity are only toxic after in vitro solubilization, probably because the protoxin is insoluble in the acidic pH of coleopteran insects. However, more recent studies show that Cry7Ab2 and Cry7Aa2 proteins solubilize in the midgut fluids of *Henosepilachna vigintioctomaculata* and *Leptinotarsa decemlineata* larvae, respectively, suggesting that the lack of solubilization involves more factors than just pH [11–13].

A large part of genetically-modified plants has Bt genes that encode proteins with insecticidal activity, with Bt cotton being the third most grown insect-resistant Genetically-modified Organism (GMO) crop in the world [14]. However, approved Bt cotton cultivars for cultivation only synthesize proteins capable of controlling insects of the Lepidoptera order, such as the cultivars Bollgard™ I (Cry1Ac), Bollgard™ II (Cry1Ac + Cry2Ab2), Bollgard™ III (cry1Ac + cry2Ab2 + vip3Aa), WideStrike™ (Cry1Ac + Cry1Fa), TwinLink™ (Cry1Ab + Cry2Ae), and VIPCOT™ (cry1Ab + vip3Aa) [15].

Several studies have reported that the widespread and almost exclusively use of *B. thuringiensis*, either as biopesticide or in transgenic plants, prioritizes a specific mode of action. This specificity leads us to the need to identify new broadly effective molecules, with different modes of action from those of *B. thuringiensis*. These molecules may interact with different receptors in the larvae intestine and safely control known and emerging agricultural pests [16].

In this sense, to diversify the use of Bt-based control methods and extend the efficient use of technology, researchers have evaluated many other biocontrol agents. Bacteria, such as *Bacillus popilliae*, *Brevibacillus laterosporus*, *Alcaligenes faecalis*, *Pseudomonas chlororaphis*, *P. mosselii*, and *Paenibacillus lentimorbus*, have been reported to express insecticidal proteins toxic to insect pests of the order Coleoptera [17–22].

The bacterial genus *Pseudomonas* has gained notoriety in the current scenario of biological control. In addition to its good track record in agriculture, this group produces the growth hormone indoleacetic acid (IAA), is used as a bioremediator, and applied to control certain fungal plant pathogens by producing phenazine-type antibiotics [23–27]. Recently, certain species have been reported as entomopathogenic and used as promising sources of insecticidal genes to produce genetically-modified plants that express insecticidal proteins against Coleoptera insects, including IPD072Aa from *P. chlororaphis* and PIP-47Aa from *P. mosselii* [20, 21]. The discovery of new protein toxins isolated from *Pseudomonas* spp., which have demonstrated efficiency in the biological control of insects, reveals potential mechanisms of action and alternatives for managing the possible emergence of insect pest resistance. This expands the gene pyramiding in genetically-modified plants, increasing the durability of protein insecticidal efficiency [28].

Due to the significant damage by *A. grandis* in cotton crops and the insecticidal potential of *Pseudomonas* spp. proteins against coleopterans, the present study aimed to evaluate the toxicity of IPD072Aa and PIP-47Aa proteins for cotton boll weevil control. We also intended to investigate the potential synergy of these proteins with Cry proteins from *B. thuringiensis* for the construction of pyramidal cotton plants, what could increase the durability of the insecticidal efficiency of the proteins and consequently the management and control of this pest.

## Materials and methods

### Insect population

Pathogenicity bioassays using cotton boll weevil larvae, *Anthonomus grandis* (Coleoptera: Curculionidae), were developed in collaboration with the Laboratory of Insect Biology of the Department of Entomology and Acarology, ESALQ-USP, in Piracicaba-SP (Brazil). The laboratory has established and maintained *A. grandis* populations for over 10 years, and it has been renewed annually with field populations. The laboratory also provided the insect's diet and eggs.

### Protein sources and preparations

Recombinant clones of *E. coli* BL21(DE3) expressing the *Pseudomonas* genes IPD072Aa (GenBank accession number KT795291) and PIP-47Aa (GenBank accession number KY982916) were synthesized by GenOne Biotechnologies (Rio de Janeiro, Brazil). After the synthesis, the genes were cloned into the pET SUMO expression vector (Invitrogen™).

The clones of *E. coli* BL21(DE3) expressing a single protein Cry1Ia10 and Cry8B from *Bacillus thuringiensis* were cloned into the pET SUMO expression vector (Invitrogen™) [29]. The *Cry3Aa* gene was obtained from the Bacillus Genetic Stock Center (BGSC) [30], subcloned into the pET SUMO expression vector (Invitrogen™), and inserted into *E. coli* BL21(DE3).

Protein expression of *Pseudomonas* spp. and *B. thuringiensis* was performed according to [31] protocol, with alteration of the isopropyl-β-D-1-thiogalactopyranoside (IPTG) quantity to a final concentration of 1 mM and incubation temperature of 22ºC.

The protein expression of each clone was verified by sodium dodecyl sulfate-polyacrylamide gel electrophoresis (SDS-PAGE). The protein concentration in each preparation was determined by densitometry of SDS-PAGE gels, using bovine serum albumin (BSA) as a standard and the ImageQuant TL 8.1 software (GE Healthcare Bio-Sciences AB, Uppsala, Sweden).

### Protein purification

Recombinant proteins were purified in 1-mL "HisTrapTM HP" columns (GE Healthcare Bio-Sciences AB, Upsala Sweden), which allow purification by affinity chromatography on "Ni

Sepharose" resin, enhancing purification of histidine-tagged proteins such as the ones studied, which were fused to a six-histidine tail (6xHis).

Before the purification, protein lysates were filtered through 0.22-μm filters. Then, they were purified on a column pre-equilibrated with an equilibration buffer (20 mM sodium phosphate pH 7.4). Afterward, the proteins were eluted with an equilibration buffer supplemented with imidazole at concentrations of 10, 25, 50, 75, and 250 mM. Finally, fractions were collected for subsequent quantification, as described in the previous section.

## Amicon ultra 30 kDa filtration

A 30 kDa mesh Amicon™ Ultra-15 filter (Millipore, Germany) was used to remove imidazole from proteins after purification. Aliquots of 15 mL purified proteins were added to concentrator tubes to be centrifuged (4000 x g; 20 min; room temperature). After the centrifugation, 15 mL of 100 mM sodium phosphate buffer pH 7.0 were added to the tubes, which were again centrifuged (4000 x g; 20 min; room temperature). To remove proteins from the filter, 7.5 mL of the same buffer was added. After the filtration, 10 mL aliquots of the samples and 100 mM sodium phosphate buffer pH 7.0 were lyophilized for 24 hours in a "Savant Super Modulyo" lyophilizer and resuspended in 1 mL of deionized water for subsequent quantification, as described in the previous section.

## *Anthonomus grandis* bioassays to estimate $LC_{50}$ and $LC_{90}$

Toxicity was assessed by incorporating the purified toxins into an artificial diet, which was dispensed in 128-well polystyrene plates ("Cell Wells, Corning Glass Works", Corning, NY). The artificial diet was prepared for *Anthonomus grandis* according to [32]. Different concentrations of *Pseudomonas* spp. and *B. thuringiensis* protein lysates were gradually increased according to [33]. They were then diluted to 7 concentrations, ranging from 2 to 256 μg mL$^{-1}$. A single *A. grandis* neonate larva was added to each well. Each plot consisted of 16 neonate larvae, with three replicates per concentration, totaling 48 larvae/concentration. These plates were sealed with a film, and a small hole was pierced in each well. Deionized water was used as a negative control for natural mortality. The trays were kept in a climatized room at 25°C (± 2°C), with a relative humidity of 70% (± 10%) and a photoperiod of 14:10 h (light: dark). Mortality was recorded after seven days of bioassay implantation, and larvae weighing <0.2 mg and not beyond the second instar were considered dead. The Polo-Plus software (LeOra Software, Berkeley, CA, USA) was used to estimate $LC_{50}$ and $LC_{90}$ in concentration-response bioassays by Probit analysis [34] and to obtain the slope values and chi-square. Differences were considered significant if the $LC_{50}$ and $LC_{90}$ estimates were not within the confidence intervals.

## Assay of different protein combinations against *Anthonomus grandis*

Observed and expected values were compared simultaneously for combinations between *Pseudomonas* spp. and *B. thuringiensis* proteins, namely: IPD072Aa/PIP-47Aa, Cry1Ia10/Cry3Aa, Cry1Ia10/Cry8B, Cry3Aa/Cry8B, IPD072Aa/Cry1Ia10, IPD072Aa/Cry3Aa, IPD072Aa/Cry8B, PIP-47Aa/Cry1Ia10, PIP-47Aa/Cry3Aa, and PIP-47Aa/Cry8B. An initial test for interactions between proteins was performed at a single concentration of each protein. In the mixture, the concentration of each toxin was selected so that it was at its respective $LC_{50}$ value. The expected mortality in absence of interactions was estimated assuming the hypothesis of simple independent action [34]. Under this hypothesis, the proportion (P) of larvae dying from

exposure to a mixture of two toxins was calculated as:

$$P = 1 - (1 - P_1)(1 - P_2) \tag{1}$$

Wherein: $P_1$ and $P_2$ represent the proportions of larvae killed by toxins 1 and 2, respectively. This formula is equivalent to equation 11.33 [34]. The observed mortality values obtained at the theoretical $LC_{50}$ value with single toxins were used to calculate the expected mortality of toxin mixtures. The significance of deviations between expected and observed mortality was determined using Fisher's exact test.

A second test for interactions was conducted using concentration-response assays, in which the proportions of two proteins in the mixture corresponded to the ratio of their respective $LC_{50}$ values. The expected mortality in absence of interactions was estimated assuming the hypothesis of simple similar action [34], using the formula [35], which derives from equation 11.8 [34]:

$$CL_{50}(m) = \frac{1}{\frac{ra}{Cl_{50(a)}} + \frac{rb}{Cl_{50(b)}}} \tag{2}$$

Wherein: $LC_{50}(m)$ is the lethal concentration of the mixture, $LC_{50}(a)$ and $LC_{50}(b)$ are the respective lethal concentrations of each component, and *ra* and *rb* are the relative proportions of the components *a* and *b* in the mixture.

After estimating the $LC_{50}(m)$, the ratio of expected $LC_{50}$ to observed $LC_{50}$ [$LC_{50}(exp)$/$LC_{50}(obs)$] was calculated to determine the interaction of the combinations of toxins. The ratio between $LC_{50}$ values indicates when the synergism factor (SF) is present in a synergistic interaction concerning the toxins in combination. Thus, an FS value > 1 indicates the occurrence of synergism between toxins, FS < 1 indicates an antagonistic interaction, and FS = 1 indicates additive toxicity [36].

## Results

### Expression of proteins

The recombinant *Pseudomonas* spp. proteins IDP072Aa and PIP-47Aa were detected through bands of molecular weight of 24 kDa and 46 kDa, respectively. The *B. thuringiensis* proteins Cry1Ia10, Cry3Aa, and Cry8B were confirmed by bands of molecular weights of about 94 kDa, 88 kDa, and 143 kDa, respectively (Fig 1). For each protein, a 13 kDa molecular weight referring to the 6-histidine tag (6×His) and SUMO protein was added, both added by the pET SUMO expression vector in the N-terminal region of the protein.

### Insecticidal activity against *Anthonomus grandis*

The concentration-response bioassays allowed us to estimate the $LC_{50}$ and $LC_{90}$ values of the five proteins (Table 1). Toxicity against *A. grandis* neonate larvae varied considerably with the protein tested. The $LC_{50}$ values ranged from 6.35 to 17.71 µg mL$^{-1}$, while $LC_{90}$ ones varied from 36.12 to 85.74 µg mL$^{-1}$. At the $LC_{50}$ level, *Pseudomonas* spp. proteins showed lower toxicity against *A. grandis* larvae compared to *B. thuringiensis* proteins. The Bt proteins Cry1Ia10 and Cry3Aa exhibited intermediate and similar toxicity to the cotton boll weevil with comparable values. However, the protein Cry8B stood out from the others for its high toxicity to neonate larvae of *A. grandis* at the $LC_{50}$ level. At the $LC_{90}$ level, only PIP-47Aa and Cry8B proteins showed significant differences, with Cry8B being more promising as it was approximately three times more toxic to *A. grandis* compared to PIP-47Aa protein from *Pseudomonas* spp. The Cry1Ia10 protein demonstrated moderate toxicity against the insect pest larvae with

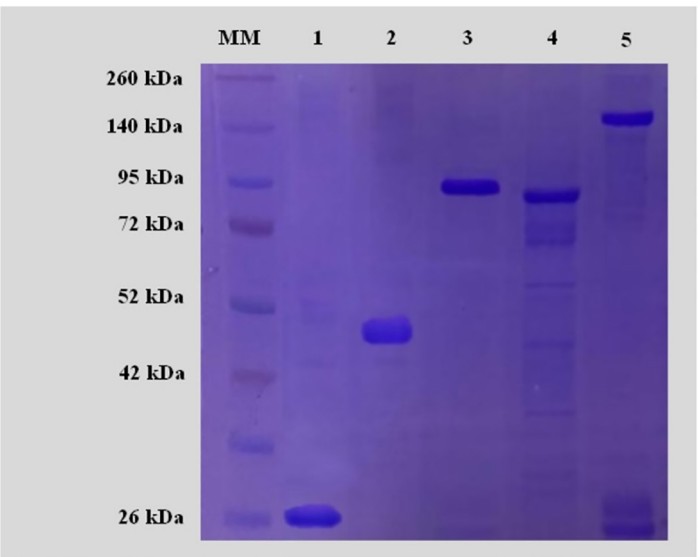

**Fig 1. Detection of purified *Pseudomonas* spp. and *Bacillus thuringiensis* proteins on a 12% SDS-PAGE gel.**
MM = molecular marker "Spectra™ Multicolor Broad Range Protein Ladder", 1 = IPD072Aa, 2 = PIP-47Aa,
3 = Cry1Ia10, 4 = Cry3Aa, 5 = Cry8B.

intermediate $LC_{50}$ values. However, at the $LC_{90}$ level, the Cry1Ia10 protein did not show significant differences from IPD072Aa and Cry3Aa proteins, indicating that both proteins are equally effective in controlling the boll weevil. Such differences at $LC_{50}$ and $LC_{90}$ may be due to the difference in the respective regression line slopes between the proteins. Negative controls did not cause mortality under the assay conditions.

## Effect of combinations between toxins on mortality rate

Several combinations of *Pseudomonas* spp. and *Bacillus thuringiensis* proteins were tested at a single concentration (corresponding to their respective $LC_{50}$ values). The observed mortality was compared to the expected mortality, assuming no interaction (Table 2). A significant interaction was found for the combination of Cry1Ia10 and Cry8B ($p < 0.05$). In this case, the interaction was negative (i.e., antagonistic) as the observed mortality was lower than expected. For the combination IPD072Aa and Cry8B, additive toxicity was observed, with the observed

**Table 1. Estimation of lethal concentrations ($LC_{50}$ and $LC_{90}$) for *Pseudomonas* spp. and *Bacillus thuringiensis* proteins against *Anthonomus grandis* neonate larvae.**

| Protein | $LC_{50}$ (CI min.-max.)[a] | $LC_{90}$ (CI min.-max.)[a] | (b ± SE)[b] | ($X^2$)[c] |
|---|---|---|---|---|
| IPD072Aa | 14.24 (8.38–21.53)b | 70.75 (43.13–171.13)ab | 1.89±0.18 | 10.2 |
| PIP-47Aa | 17.71 (13.94–22.01)b | 85.74 (63.83–128.15)b | 1.87±0.19 | 2.39 |
| Cry1Ia10 | 10.16 (6.37–14.51)ab | 52.80 (34.53–105.76)ab | 1.69±0.19 | 7.08 |
| Cry3Aa | 7.82 (5.57–10.19)ab | 46.93 (34.59–71.65)ab | 1.64±0.18 | 4.57 |
| Cry8B | 6.35 (4.40–8.36)a | 36.12 (26.78–55.15)a | 1.19±0.16 | 0.76 |

[a] Values expressed as µg mL$^{-1}$ at a confidence interval (CI) of 95%. Values followed by the same lowercase letter in the same column do not differ from each other at $p > 0.05$;

[b] Angular coefficient of the line and standard error;

[c] Chi-square ($p > 0.05$).

**Table 2. Mortality of *Anthonomus grandis* neonate larvae due to the application of *Pseudomonas* spp. and *Bacillus thuringiensis* proteins analyzed separately and in combination.**

| Protein/Combination | (Concentration)[a] | Mortality (%) | | Fisher's Exact Test[d] | X² (p)[e] |
|---|---|---|---|---|---|
| | | (Observed)[b] | (Expected)[c] | | |
| IPD072Aa | 14.24 | 52 | 50 | | |
| PIP-47Aa | 17.71 | 48 | 50 | | |
| Cry1Ia10 | 10.16 | 46 | 50 | | |
| Cry3Aa | 7.82 | 50 | 50 | | |
| Cry8B | 6.35 | 54 | 50 | | |
| IPD072Aa/PIP-47Aa | 14.24+17.71 | 79 | 75 | 0.4043 | 0.2360 (0.6272) |
| Cry1Ia10/Cry3Aa | 10.16+7.82 | 81 | 73 | 0.2334 | 0.9430 (0.3314) |
| Cry1Ia10/Cry8B | 10.16+6.35 | 40 | 75 | 0.0004** | 12.303 (0.0005**) |
| Cry3Aa/Cry8B | 7.82+6.35 | 81 | 77 | 0.4011 | 0.2530 (0.6152) |
| IPD072Aa/Cry1Ia10 | 14.24+10.16 | 79 | 74 | 0.3166 | 0.5150 (0.4732) |
| IPD072Aa/Cry3Aa | 14.24+7.82 | 79 | 76 | 0.4043 | 0.2360 (0.6272) |
| IPD072Aa/Cry8B | 14.24+6.35 | 77 | 78 | 0.5957 | 0.0000 (1.0000) |
| PIP-47Aa/Cry1Ia10 | 17.71+10.16 | 75 | 71 | 0.4093 | 0.2110 (0.6460) |
| PIP-47Aa/Cry3Aa | 17.71+7.82 | 77 | 74 | 0.4070 | 0.2220 (0.6374) |
| PIP-47Aa/Cry8B | 17.71+6.35 | 77 | 76 | 0.5000 | 0.0570 (0.8110) |

[a] Protein concentrations were chosen to match their respective $LC_{50}$ values and expressed as µg mL$^{-1}$;

[b] Each value represents the average of three replicates of 16 larvae per replicate (n = 48);

[c] Expected mortality considering simple and independent action;

[d] Values where q p>0.05 indicate non-significant differences;

[e] Chi-square and *p* values.

mortality being equal to the expected mortality. Therefore, the resulting additivity of the combined protein application produced a toxic effect on *A. grandis* neonate larvae similar to their effect when applied alone.

## Analysis of synergic effects in *Anthonomus grandis* by concentration-response assays

The combinations used for mortality analysis were further investigated by concentration-response assays. Synergism was observed in most of the combinations tested, except for Cry1Ia10 with Cry8B and IPD072AA with Cry8B. Mild synergism was observed in two combinations, IPD072Aa with Cry1Ia10 and PIP-47Aa with Cry8B (FS = 1.09 and 1.02, respectively), whereas moderate synergism was observed in five protein combinations (FS = 1.12 to 1.41). The combination of Cry1Ia10 with Cry3Aa showed a high synergistic interaction (FS = 2.06) (Table 3).

The combination of IPD072Aa with Cry8B had an expected $LC_{50}$ within the range for this mixture, hence considered an additive interaction (FS = 1). Therefore, the final effect of both combined proteins is equal to the sum of their individual effects. The observed $LC_{50}$ of the combination of Cry1Ia10 with Cry8B was higher than expected, demonstrating antagonism for this interaction.

## Discussion

Both *Pseudomonas* spp. and *Bacillus thuringiensis* proteins presented sizes compatible with those already reported, plus 13 kDa relative to the 6-histidine tag (6xHis) and SUMO protein,

**Table 3. Interactions between *Pseudomonas* spp. and *Bacillus thuringiensis* proteins for control of *Anthonomus grandis* neonate larvae.**

| Protein combination | $LC_{50}$ (CI min.-max.)[a] Observed | $LC_{50}$ (Expected)[b] | $SF^c$ | $(b \pm SE)^d$ | $(X^2)^e$ |
|---|---|---|---|---|---|
| IPD072Aa/PIP-47Aa (1:1) | 6.82 (3.32–10.59) | 7.89 | 1.16 | 1.74±0.20 | 8.70 |
| Cry1Ia10/Cry3Aa (1:1) | 2.15 (0.69–3.99) | 4.42 | 2.06 | 1.06±0.18 | 1.26 |
| Cry1Ia10/Cry8B (1:1) | 6.41 (4.28–8.62) | 3.91 | 0.61 | 1.60±0.20 | 4.19 |
| Cry3Aa/Cry8B (1:1) | 2.48 (1.03–3.98) | 3.50 | 1.41 | 1.51±0.27 | 0.71 |
| IPD072Aa/Cry1Ia10 (1:1) | 5.45 (3.17–8.13) | 5.93 | 1.09 | 2.15±0.27 | 3.92 |
| IPD072Aa/Cry3Aa (1:1) | 4.28 (3.12–5.49) | 5.05 | 1.18 | 1.83±0.25 | 2.98 |
| IPD072Aa/Cry8B (2:1) | 3.36 (2.41–4.28) | 3.36 | 1.00 | 2.01±0.28 | 2.20 |
| PIP-47Aa/Cry1Ia10 (1:1) | 5.19 (3.98–6.49) | 6.46 | 1.24 | 2.02±0.26 | 1.19 |
| PIP-47Aa/Cry3Aa (2:1) | 3.72 (2.74–4.70) | 4.15 | 1.12 | 1.93±0.26 | 1.29 |
| PIP-47Aa/Cry8B (2:1) | 3.64 (2.64–4.65) | 3.70 | 1.02 | 1.85±0.25 | 2.84 |

[a] Values expressed in µg mL-1 with a 95% confidence interval (CI);

[b] Expected mortality assuming simple similar action;

[c] SF: Synergism Factor, calculated as the ratio of expected $LC_{50}$ over observed $LC_{50}$;

[d] Slope coefficient and standard error;

[e] Chi-square (p>0.05).

both referring to the pET SUMO expression vector. The Cry1Ia10 protein expression observed in this study (Fig 1) is consistent with the findings [28, 29]. For Cry3Aa protein, an 88-kDa band was obtained, which agrees with a previous report in the literature ranging from 73 to 75 kDa [37]. The Cry8B protein was confirmed by a 143 kDa band (Fig 1), following the values previously described in the literature, which range from 128 to 137 kDa for Cry8 proteins [38–40]. [20, 21] reported an 11 kDa band size for the IPD072Aa protein and 32 kDa for the PIP-47Aa protein from *Pseudomonas* spp., which agrees with our results.

Bioassays to determine the potency of proteins with potential insecticidal activity, such as *Bacillus thuringiensis* and *Pseudomonas* spp., are typically performed with individual proteins. These tests are useful to estimate the theoretical contribution of proteins to pest control when used in biological insecticides or expressed in Bt crops. However, some combinations of insecticidal proteins are known to have potential synergistic or antagonistic effects [29, 41, 42]. Therefore, when selecting combinations of genes encoding insecticidal proteins to be expressed in plants, not only differences in their mode of action should be considered, but also potential interactions between them.

Insects are known to exhibit varying degrees of susceptibility to various insecticidal toxins, whether from *B. thuringiensis* or *Pseudomonas* spp.; therefore, such susceptibility must be analyzed before commercial cultivation of genetically-modified crops. [43] noted that strains harboring genes encoding proteins belonging to the Cry1, Cry3, Cry5, Cry7, Cry8, Cry9, Cry10, Cry14, Cry18, Cry22, Cry34, Cry35, and Cry36 classes exhibit insecticidal activity against coleopterans insects. In our study, the toxicity levels of *B. thuringiensis* and *Pseudomonas* spp. proteins against *A. grandis* were evaluated, with $LC_{50}$ and $LC_{90}$ ranging from 6.35 to 17.71 and 36.12 to 85.74 µg mL$^{-1}$, respectively (Table 1). Among the proteins tested, Cry8B was the most active against the pest. At $LC_{50}$, this protein showed significant differences and was two and three times more toxic than the toxins IPD072Aa and PIP-47Aa, respectively. Meanwhile, the proteins Cry1Ia10 and Cry3Aa were equally toxic to the insect larvae. At the $LC_{90}$ level, the Cry8B protein proved to be more active than the PIP-47Aa protein, which showed a higher $LC_{90}$ value. While the IPD072Aa, Cry1Ia10, and Cry3Aa proteins were equally toxic to *A. grandis* larvae.

When the results of $LC_{50}$ and $LC_{90}$ were compared, differences occurred due to higher slopes in concentration-mortality regression lines among Cry8B proteins (1.19); between Cry3Aa and Cry1Ia10 (1.64 and 1.69); between IPD072Aa and PIP-47Aa (1.87 and 1.89) (Table 1). According to the $LC_{50}$ and $LC_{90}$ values, *Pseudomonas* spp. proteins required higher concentrations to be effective against the pest insect, but once a critical threshold is reached, the response increases rapidly with concentration. These high slopes for IPD072Aa and PIP-47Aa proteins may be associated with the fact that this type of protein requires a concentration threshold in the midgut of the insect. By contrast, Bt proteins showed a more common concentration-mortality response, represented by a shallower slope.

In the current study, the insecticidal proteins isolated from *B. thuringiensis*, Cry1Ia10, Cry3Aa, and Cry8B, were shown to be efficient in controlling *A. grandis* larvae, with low $LC_{50}$ estimates of 10.16, 7.82, and 6.35 µg mL$^{-1}$, respectively (Table 1). Among the reported Bt proteins with activity against coleopteran insects, those from the Cry3 and Cry8 families have a broader spectrum of susceptible insects [44]. In the Cry3 group of proteins, such as Cry3Aa, Cry3Ba, Cry3Bb, and Cry3Ca, they showed activity against most major coleopteran families, including Chrysomelidae, Curculionidae, Scarabaeidae, and Tenebrionidae [44]. On the other hand, Cry8-type proteins, such as Cry8A and Cry8B, exhibited activity against the chrysomelid beetles *L. decemlineata* and *Diabrotica* spp., Cry8Ca against the tenebrionid beetle *Alphitobius diaperinus*, Cry8Ea and Cry8Na1 against *Holotrichia parallela* larvae, Cry8Ga1 against *H. parallela* and *H. oblita*, and Cry8Ka against the weevil *A. grandis* [38, 39, 45–47]. These results confirm the effectiveness and specificity of the *Cry3* and *Cry8* gene groups in controlling coleopteran agricultural pests.

In previous studies on the toxicity of Cry1Ia protein against *A. grandis* and *Spodoptera frugiperda*, purified recombinant Cry1Ia12 protein showed to be toxic to the larvae of both insects, with the highest concentrations tested to achieve maximum toxicity being 230 µg mL$^{-1}$ and 5 µg mL$^{-1}$, respectively [48]. This concentration is 10 times less toxic than that found by [49] for Cry1Ia proteins from *B. thuringiensis* in a recombinant baculovirus system against *A. grandis* and *S. frugiperda*. In both cases, the estimated concentrations for Cry1I protein were less efficient than those found in our study (Table 1).

Assays conducted with the Cry1Ia protein against neonate larvae of *A. grandis* and *S. frugiperda* demonstrate to be a viable alternative for cotton pest control. These results were favorable for [50] experimentally generated a genetically-modified cotton plant capable of expressing the *Bt Cry1Ia10* gene, which has dual action and efficiently controls insects of the Lepidoptera and Coleoptera orders, thus controlling the main pests of the crop. Although some *B. thuringiensis* genes can produce lethal toxins against some pest insects, commercially available genetically modified cotton plants resistant to *A. grandis* do not yet exist [51]. These data demonstrate the high importance of studying and finding new genes to expand the design of new combinations of pyramided genes in genetically modified crops for the control of the cotton boll weevil.

Searching for biopesticides or alternative approaches, such as attract-and-kill strategies, is of utmost importance. Allied with that, new protein toxins have been developed from different *Pseudomonas* species. Some of these proteins, such as IPD072Aa and PIP-47Aa, do not match any other protein amino acid sequence currently deposited in databases, so they are specifically toxic to Coleoptera insects in their monomeric form [52]. [21] reported an $LC_{50}$ of 52.5 µg mL$^{-1}$ for Western Corn Rootworm (*Diabrotica virgifera virgifera*) neonate larvae after four days of exposure to the insecticidal protein PIP-47Aa from *P. mosselii*. [20] estimated an $LC_{50}$ of about 120 µg mL$^{-1}$ for IPD072Aa protein from *P. chlororaphis* after eight days of exposure. In our study, we report a lower $LC_{50}$ for seven days of exposure against neonate larvae of the coleopteran *A. grandis*, with PIP-47Aa being about three times more toxic (17.71 µg mL$^{-1}$),

and IPD72Aa being 8 times more toxic (14.24 µg mL$^{-1}$), as in Table 1. These results highlight the importance of finding new genes to increase the number of insect pests controlled by the action of different proteins with insecticidal potential.

Understanding the effects of protein interactions is crucial when selecting toxins for pest management and control. To determine the efficacy of a mixture, it is important to consider not only the effectiveness of its components but also the synergy between them. If the mixture is more toxic than expected, it suggests a synergistic interaction between the components, while if it is less toxic than expected, an antagonistic interaction is implied [34].

Numerous studies have investigated the interactions between Bt proteins in different insect species. For instance [29], revealed that the combination of Vip3Aa and Cry1Ia proteins exhibited synergy against lepidopteran pests such as *S. frugiperda*, *S. albula*, and *S. cosmioides* while demonstrating slight antagonism in *S. eridania*. Another study found a slight synergistic effect of combining Vip3Aa and Cry9Ca in *Plutella xylostella* [41]. Synergistic and antagonistic interactions between Vip3Aa and Cyt2Aa proteins have been reported in *Chilo suppressalis*, *Spodoptera exigua*, *Chironomus tepperi*, *Helicoverpa armigera*, and *Culex quinquefasciatus* [42]. [53] tested combinations of Cry and Vip proteins from *B. thuringiensis* and reported antagonism in all combinations tested against *S. frugiperda*, but synergy in *Diatraea saccharalis* with the combination of Vip3Aa and Cry1Ca. [54] investigated interactions between other Cry1, Cry2, and Vip3 proteins against *Spodoptera frugiperda* larvae and found synergistic interactions in all combinations tested. These findings underscore the significance of exploring interactions between different proteins in the control and management of insect pests.

In this study, we tested various combinations of proteins to identify potential interactions among them. The combination of Cry1Ia10 and Cry3Aa produced the highest synergistic effect (SF = 2.06) (Table 3). It has been proven that the Cry1Ia protein is toxic to *A. grandis* [48, 49], and strains of *Bt* subsp. *tenebrionis* containing the Cry3 gene have been described as the Cry protein group that shows the most activity against the majority of coleopteran species under study [44, 55]. Therefore, the combination of Cry1Ia10 and Cry3Aa could be used in cotton to achieve broader control of this target species, which would aid in management the possible emergence of pest resistance, as there are already reports that the widespread use of Cry protein-based insecticides and Bt crops carries the risk of selecting tolerant insect biotypes, such as the emergence of resistant populations of chrysomelid beetles *Leptinotarsa decemlineata*, *Chrysomela scripta* under laboratory conditions, and *Diabrotica* spp. to Bt corn [56–58].

We investigated potential interactions between Cry3Aa and PIP-47Aa proteins when combined with other proteins. Clear synergistic effects were evidenced by the substantial differences between expected and observed LC$_{50}$ values (Table 3). Several hypotheses have been proposed to explain the mechanisms underlying this synergy. One such hypothesis is that the proteins can form hetero-oligomers that are more efficient at inserting into the membrane than their corresponding homo-oligomers, resulting in greater toxicity against target pests [59]. Another hypothesis is that combinations of two proteins can induce the formation of larger pores in the larval midgut membrane than when each protein acts alone. In support of this idea [60], demonstrated that a combination of Cry1 toxins exhibited synergistic activity against *Lymantria dispar* caterpillars, with the combination leading to the formation of larger pores than when the toxins were used individually.

The synergistic interactions observed in our study (Table 3) suggest that a pyramided cotton plant expressing these protein combinations could be a valuable addition to overall pest management programs for *A. grandis*. In a related study [61], reported on a genetically-modified cotton plant with insecticidal activity against *A. grandis* conferred by the Cry10Aa toxin. Susceptibility bioassays involving feeding *A. grandis* on GM cotton leaves and floral buds

demonstrated a significant entomotoxic effect and a high level of insect mortality, reaching up to 100% efficiency.

In addition to discovering *B. thuringiensis*-isolated molecules for effective control of agricultural pests, researchers have also developed transgenic plants with efficient *Pseudomonas* spp. genes for controlling Coleopteran pests. For instance, the gene IPD072Aa from *Pseudomonas chlororaphis*, which encodes the IPD072Aa protein, was utilized in the genetic transformation of corn plants after its insecticidal activity was confirmed. The resulting plants were found to be resistant to *Diabrotica virgifera virgifera* [20]. [21] isolated the PIP-47Aa protein from *Pseudomonas mosselii*, which demonstrated insecticidal activity against *Diabrotica virgifera virgifera*, *Diabrotica barberi*, *Diabrotica undecimpunctata howardi*, *Diabrotica speciosa*, and *Phyllotreta cruciferae*, and was also effective when expressed in corn plants.

In both studies, these new proteins remained effective against *Diabrotica virgifera virgifera* larvae resistant to the mCry3Aa protein, as well as the binary insecticidal protein Cry34Ab1/Cry35Ab1 (recently renamed Gpp34Ab1/Tpp35Ab1 under a revised nomenclature system [62]), both used in commercial events of transgenic corn, which puts us in front of potential mechanisms of action and new alternatives for insect control management, as pyramiding of multiple new insecticidal proteins with distinct target sites in the midgut of insects is also essential to increase the durability of the technology.

In this study, synergistic effects were observed for most combinations of *Pseudomonas* spp.-isolated proteins tested, except for the combination between IPD072Aa and Cry8B, which showed an additive effect (Tables 2 and 3). This indicates that the effect of this combination is equal to the sum of the individual effects of each protein. An antagonistic effect between the Cry1Ia10 and Cry8B toxins in *A. grandis* was also observed (Tables 2 and 3). This may be due to physical interactions between the two proteins that render them inactive. Alternatively, complex formation may simply mask an epitope on the more toxic protein, preventing it from interacting with the membrane receptor. Antagonism can also result from steric interactions, where both toxins bind to different epitopes on the same membrane molecule [60, 63]. Further studies are necessary to determine the reason for the antagonistic effect between Cry1Ia10 and Cry8B observed in *A. grandis*.

As combinations of *Pseudomonas* spp. and *Bacillus thuringiensis* proteins can result in synergistic, antagonistic, or additive interactions, different biochemical modes of action may be involved in controlling the studied insect species. However, our results were obtained under laboratory conditions, where predators and parasitoids were absent, and an artificial diet was used without phytochemicals. Therefore, to confirm whether the interactions found in the study persist or even increase in the field, protein combinations that exhibited interactions under laboratory conditions should be tested under field conditions, where both proteins will be expressed in the same genetically-modified plant.

## Conclusion

The results presented in this study showed that proteins from *Pseudomonas* spp. and *Bacillus thuringiensis* were efficient in controlling neonate larvae of *A. grandis*, both individually and in most of the tested combinations. These data point to the need to explore microbial biodiversity in the search for new proteins and mechanisms of action to diversify the use of a single control method, increase the efficiency of agricultural pest control, and manage the potential emergence of insect resistance, favoring the extended use of technology at the field level. Due to the combinations of toxins from *Pseudomonas* spp. and *B. thuringiensis* against *A. grandis* showing synergistic, antagonistic, and additive interactions, it suggests that there are different types of interactions within the host, and consequently, different modes of action of the tested proteins.

It becomes clear that in future perspectives, research should be conducted through binding assays to elucidate the mode of action of different toxins, in order to identify specific membrane receptors and the possibility of competition for the same binding site among the proteins.

## Supporting information

**S1 File.**
(DOCX)

**S1 Raw images.**
(DOCX)

## Acknowledgments

The authors thank the Prof. Dr. José Roberto Postali Parra, technician Neide Graciano Zério, from the Laboratory of Insect Biology in the Department of Entomology and Acarology of the ESALQ/USP, for kindly providing eggs of the species evaluated in this study and São Paulo State University (UNESP), Faculty of Agrarian and Veterinary Sciences (Jaboticabal Campus), Biology Department.

## Author Contributions

**Conceptualization:** Jardel Diego Barbosa Rodrigues, Janete Apparecida Desidério.

**Data curation:** Jardel Diego Barbosa Rodrigues, Jackson Antônio Marcondes de Souza.

**Formal analysis:** Jardel Diego Barbosa Rodrigues, Raquel Oliveira Moreira, Jackson Antônio Marcondes de Souza, Janete Apparecida Desidério.

**Investigation:** Jardel Diego Barbosa Rodrigues.

**Methodology:** Jardel Diego Barbosa Rodrigues, Raquel Oliveira Moreira.

**Project administration:** Jardel Diego Barbosa Rodrigues.

**Resources:** Jardel Diego Barbosa Rodrigues, Janete Apparecida Desidério.

**Supervision:** Jackson Antônio Marcondes de Souza, Janete Apparecida Desidério.

**Validation:** Janete Apparecida Desidério.

**Visualization:** Jardel Diego Barbosa Rodrigues.

**Writing – original draft:** Jardel Diego Barbosa Rodrigues, Raquel Oliveira Moreira.

**Writing – review & editing:** Jardel Diego Barbosa Rodrigues, Raquel Oliveira Moreira, Jackson Antônio Marcondes de Souza, Janete Apparecida Desidério.

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
