## [Decision Letter · Decision Letter 0]

7 Jun 2023

PONE-D-23-14450Interaction of insecticidal proteins from Bacillus thuringiensis and Pseudomonas spp. for boll weevil resistance managementPLOS ONE

Dear Dr. Barbosa Rodrigues,

Thank you for submitting your manuscript to PLOS ONE. After careful consideration, we feel that it has merit but does not fully meet PLOS ONE’s publication criteria as it currently stands. Therefore, we invite you to submit a revised version of the manuscript that addresses the points raised during the review process.

We look forward to receiving your revised manuscript.

Kind regards,

Md. Asaduzzaman Shishir, PhD

Academic Editor

PLOS ONE

Journal Requirements:

5. Please remove your figures from within your manuscript file, leaving only the individual TIFF/EPS image files, uploaded separately. These will be automatically included in the reviewers’ PDF.

Reviewers' comments:

Reviewer's Responses to Questions

**Comments to the Author**

1. Is the manuscript technically sound, and do the data support the conclusions?

Reviewer #1: Partly

Reviewer #2: Partly

2. Has the statistical analysis been performed appropriately and rigorously? 

Reviewer #1: I Don't Know

Reviewer #2: No

3. Have the authors made all data underlying the findings in their manuscript fully available?

Reviewer #1: No

Reviewer #2: Yes

4. Is the manuscript presented in an intelligible fashion and written in standard English?

Reviewer #1: Yes

Reviewer #2: No

5. Review Comments to the Author

Reviewer #1: There is a discussion but no conlusion. Otherwise the manuscript is sound. A conclusion with future direction(s) must be included in the manuscript before publication. There is few minor gramtical mistakes. Those also should be addressed.

Reviewer #2: The manuscript titled "Interaction of insecticidal proteins from Bacillus thuringiensis and Pseudomonas spp. for boll weevil resistance management" describes the expression of toxins from B. thuringiensis and Pseudomonas spp., their isolated and combined activities against the cotton boll weevil (Anthonomus grandis). Despite the significance of finding new molecules to control this important pest, I have several concerns about the manuscript, and unfortunately, it cannot be accepted in its present form.

Major Issues:

1. In the abstract and introduction, the authors have raised several confusions regarding the boll weevil. Firstly, A. grandis is the most important cotton pest in the Americas, not worldwide. Additionally, there are no commercially available transgenic crops or pesticides based on Bt Cry toxins specifically targeting the boll weevil, so the issue of its resistance to Bt toxins does not exist. These statements need to be corrected.

2. The introduction is overly focused on Brazil, while it should be noted that A. grandis is an important pest from the US to Argentina.

3. In lines 79-80, the authors state that "Upon ingestion of these proteins, the high intestinal pH of susceptible larvae activates a protoxin that binds to specific receptors…". This statement is true only for lepidopteran insects; for other insects, such as coleopterans, the mechanism is slightly different. This distinction should be clarified in the text.

4. The statistical analysis of Table 1 data is misinterpreted. The authors claim certain statistical differences that are not valid. For example, at LC50, Cry1Ia10 overlaps with all the other proteins, and at LC90, only Cry8 and PIP-47 show significant differences. These errors should be corrected, along with the sentences referring to these results throughout the manuscript.

5. The discussion section needs to be reformulated as it contains excessive repetition.

6. Additionally, I believe the text would benefit from an additional round of language corrections.

Minor Issues:

Line 34: Change "Coleoptera" to "coleopteran."

Line 37: Specify which Cry8 variant was used: Cry8Aa, B, or C?

Line 83: Change "action" to "activity."

Line 101: Change "methods" to "agents."

Line 195: Change "Assay of interactions between proteins against Anthonomus grandis" to "Assay of different protein combinations against Anthonomus grandis."

Line 268: In Table 1, change "IC" to "CI."

Lines 310 and 314: Change "FS" to "SF."

Lines 327-330: This sentence should be moved to the beginning of the paragraph.

Lines 343-344: Were all these toxins expressed, purified, and tested against coleopteran insects? Or were strains that harbor these protein genes found to be toxic to coleopterans?

Line 345: Change "Coleoptera" to "coleopteran."

Lines 353-354: Correct the decimal numbers; some use periods, while others use commas as separators.

Lines 371-375: Please verify if the bioassay conditions were the same for the compared experiments.

Lines 377-378: I fail to see how this sentence contributes to the discussion. The cited reference pertains to a different toxin expressed in a different system, making it inappropriate for dose comparison.

Lines 386-387: This sentence lacks meaning.

Lines 396-399: Were the tested larvae in the same instar or size in these experiments? Larval size can influence LC values, so it is important to consider this when making comparisons.

Line 425: Cry1Ia is highly toxic to A. grandis compared to which other Cry protein? For comparison, in Lepidoptera, the LC50 values are typically in ng, indicating a much lower dosage is required to kill the insect larvae.

Line 431: The authors could further explore the competition of toxins for binding sites in the discussion.

Line 442: Put "Lymantria dispar" in italics.

Line 460: What is "Howardi" referring to?

In the citations and references, please avoid the use of Master's and PhD theses. Instead, search for the related published articles that have undergone multiple rounds of peer review.

6. PLOS authors have the option to publish the peer review history of their article (what does this mean?). If published, this will include your full peer review and any attached files.

Reviewer #1: No

Reviewer #2: No

---

## [Author Response · Author response to Decision Letter 0]

31 Aug 2023

All considerations made by the academic editor and the reviewers are described in the response letter to the reviewers.

---

## [Decision Letter · Decision Letter 1]

20 Sep 2023

PONE-D-23-14450R1Interaction of insecticidal proteins from Pseudomonas spp. and Bacillus thuringiensis for boll weevil managementPLOS ONE

Dear Dr. Barbosa Rodrigues,

Thank you for submitting your manuscript to PLOS ONE. After careful consideration, we feel that it has merit but does not fully meet PLOS ONE’s publication criteria as it currently stands. Therefore, we invite you to submit a revised version of the manuscript that addresses the points raised during the review process.

We look forward to receiving your revised manuscript.

Kind regards,

Md. Asaduzzaman Shishir, PhD

Academic Editor

PLOS ONE

Journal Requirements:

Additional Editor Comments (if provided):

The manuscript was improved significantly after the revisions. A few more adjustments are required. Please follow the reviewers instructions.

Reviewers' comments:

Reviewer's Responses to Questions

**Comments to the Author**

1. If the authors have adequately addressed your comments raised in a previous round of review and you feel that this manuscript is now acceptable for publication, you may indicate that here to bypass the “Comments to the Author” section, enter your conflict of interest statement in the “Confidential to Editor” section, and submit your "Accept" recommendation.

Reviewer #1: All comments have been addressed

Reviewer #2: (No Response)

2. Is the manuscript technically sound, and do the data support the conclusions?

Reviewer #1: Yes

Reviewer #2: Yes

3. Has the statistical analysis been performed appropriately and rigorously? 

Reviewer #1: I Don't Know

Reviewer #2: Yes

4. Have the authors made all data underlying the findings in their manuscript fully available?

Reviewer #1: Yes

Reviewer #2: Yes

5. Is the manuscript presented in an intelligible fashion and written in standard English?

Reviewer #1: Yes

Reviewer #2: Yes

6. Review Comments to the Author

Reviewer #1: (No Response)

Reviewer #2: All of the aforementioned concerns were effectively resolved. Only one last thing, the letter denoting statistical difference for Cry3Aa in table 1 should be revised from 'a' to 'ab'.

7. PLOS authors have the option to publish the peer review history of their article (what does this mean?). If published, this will include your full peer review and any attached files.

Reviewer #1: No

Reviewer #2: No

---

## [Author Response · Author response to Decision Letter 1]

3 Nov 2023

All considerations made by the academic editor and the reviewers are described in the response letter to the reviewers.

---

## [Editor Report · Decision Letter 2]

7 Nov 2023

Interaction of insecticidal proteins from Pseudomonas spp. and Bacillus thuringiensis for boll weevil management

PONE-D-23-14450R2

Dear Dr. Barbosa Rodrigues,

We’re pleased to inform you that your manuscript has been judged scientifically suitable for publication and will be formally accepted for publication once it meets all outstanding technical requirements.

Kind regards,

Md. Asaduzzaman Shishir, PhD

Academic Editor

PLOS ONE
---

## [Editor Report · Acceptance letter]

22 Nov 2023

PONE-D-23-14450R2 

Interaction of insecticidal proteins from *Pseudomonas* spp. and *Bacillus thuringiensis* for boll weevil management 

Dear Dr. Barbosa Rodrigues:

I'm pleased to inform you that your manuscript has been deemed suitable for publication in PLOS ONE. Congratulations! Your manuscript is now with our production department. 

Kind regards, 

on behalf of

Dr. Md. Asaduzzaman Shishir 

Academic Editor

PLOS ONE